# *Dinophysis acuminata* or *Dinophysis acuta*: What Makes the Difference in Highly Stratified Fjords?

**DOI:** 10.3390/md21020064

**Published:** 2023-01-19

**Authors:** Ángela M. Baldrich, Patricio A. Díaz, Gonzalo Álvarez, Iván Pérez-Santos, Camila Schwerter, Manuel Díaz, Michael Araya, María Gabriela Nieves, Camilo Rodríguez-Villegas, Facundo Barrera, Concepción Fernández-Pena, Sara Arenas-Uribe, Pilar Navarro, Beatriz Reguera

**Affiliations:** 1Programa de Doctorado en Ciencias, Universidad de Los Lagos, Camino Chinquihue Km 6, Puerto Montt 5480000, Chile; 2Centro i~mar, Universidad de Los Lagos, Casilla 557, Puerto Montt 5480000, Chile; 3CeBiB, Universidad de Los Lagos, Casilla 557, Puerto Montt 5480000, Chile; 4Facultad de Ciencias del Mar, Departamento de Acuicultura, Universidad Católica del Norte, Coquimbo 1780000, Chile; 5Centro de Investigación y Desarrollo Tecnológico en Algas (CIDTA), Facultad de Ciencias del Mar, Universidad Católica del Norte, Coquimbo 1780000, Chile; 6Centro de Investigación Oceanográfica COPAS Sur-Austral y COPAS COASTAL, Universidad de Concepción, Concepción 4030000, Chile; 7Centro de Investigaciones en Ecosistemas de la Patagonia (CIEP), Coyhaique 5950000, Chile; 8Programa de Investigación Pesquera, Instituto de Acuicultura, Universidad Austral de Chile, Sede Puerto Montt, Puerto Montt 5480000, Chile; 9Centro Austral de Investigaciones Científicas (CADIC-CONICET), Houssay 200, Ushuaia 9410, Argentina; 10Centro Oceanográfico de A Coruña, Instituto Español de Oceanografía (IEO-CSIC), 15001 A Coruña, Spain; 11Centro Oceanográfico de Vigo, Instituto Español de Oceanografía (IEO-CSIC), Subida a Radio Faro 50, 36390 Vigo, Spain

**Keywords:** *Dinophysis acuminata*, *D. acuta*, DSP toxins, PTX2, Chilean fjords, niche partitioning

## Abstract

*Dinophysis acuminata* and *D. acuta,* which follows it seasonally, are the main producers of lipophilic toxins in temperate coastal waters, including Southern Chile. Strains of the two species differ in their toxin profiles and impacts on shellfish resources. *D. acuta* is considered the major cause of diarrhetic shellfish poisoning (DSP) outbreaks in Southern Chile, but there is uncertainty about the toxicity of *D. acuminata,* and little information on microscale oceanographic conditions promoting their blooms. During the austral summer of 2020, intensive sampling was carried out in two northern Patagonian fjords, Puyuhuapi (PUY) and Pitipalena (PIT), sharing *D. acuminata* dominance and *D. acuta* near detection levels. Dinophysistoxin 1 (DTX 1) and pectenotoxin 2 (PTX 2) were present in all net tow samples but OA was not detected. Although differing in hydrodynamics and sampling dates, *D. acuminata* shared behavioural traits in the two fjords: cell maxima (>10^3^ cells L^−1^) in the interface (S ~ 21) between the estuarine freshwater (EFW)) and saline water (ESW) layers; and phased-cell division (µ = 0.3–0.4 d^−1^) peaking after dawn, and abundance of ciliate prey. Niche analysis (Outlying Mean Index, OMI) of *D. acuta* with a high marginality and much lower tolerance than *D. acuminata* indicated an unfavourable physical environment for *D. acuta* (bloom failure). Comparison of toxin profiles and *Dinophysis* niches in three contrasting years in PUY—2020 (*D. acuminata* bloom), 2018 (exceptional bloom of *D. acuta*), and 2019 (bloom co-occurrence of the two species)—shed light on the vertical gradients which promote each species. The presence of FW (S < 11) and thermal inversion may be used to provide short-term forecasts of no risk of *D. acuta* blooms and OA occurrence, but *D. acuminata* associated with DTX 1 pose a risk of DSP events in North Patagonian fjords.

## 1. Introduction

Phytoplankton growth and distribution are affected by physical, chemical, and biological processes and their interactions which occur on a wide variety of spatial and temporal scales [1]. To study these interactions, physical–chemical properties and the target organisms should be sampled with the same spatio-temporal resolution [2,3]. Semi-enclosed systems, such as fjords and coastal embayments, possess a variety of micro-environments and marked vertical and horizontal gradients that promote growth and retention of phytoplankton populations, including those of harmful microalgal species [2,4]. Vertical distributions of motile dinoflagellates (including *Dinophysis*) are largely controlled by interactions between microalgal morphology (size, shape, and colony formation) and behaviour (swimming, diurnal vertical migration, and aggregation) [5], and by physical and chemical processes on micro- and fine scales [6,7]. These include specific microalgal adaptations to changes in physical (temperature, salinity, light, buoyancy frequency, and turbulence) and chemical (dissolved oxygen and inorganic nutrients) gradients [8,9].

Dinoflagellates and ciliates are able to adjust their positions in response to environmental changes according to their physiological and life history requirements. High density populations often aggregate in “thin layers” (TLs) where environmental and physiological parameters differ significantly from those in the layers above and below [9,10,11,12,13]. *Dinophysis* species are no exception [14,15,16]. These TLs may escape detection by conventional monitoring methods [17] and progress to understand that their formation, maintenance, and dissipation requires high resolution measurements of environmental parameters (turbulence, tidal cycle, and wind stress) concurrently with monitoring of circadian rhythms [2,18].

To date, 10 species of *Dinophysis* have been found to produce one or two groups of lipophilic toxins. The first group, the okadaic acid (OA) and dinophysistoxins (DTXs), are the *bona fide* diarrhetic shellfish poisoning (DSP) toxins. The second group, co-extracted with the former, are the pectenotoxins, which are lethal to mice via intraperitoneal injection and were, until recently, regulated together with the OA derivatives in different parts of the world, including Europe [19]. PTXs are harmless via oral administration to mammals [20] although purified PTXs cause deleterious effects in early life stages of bivalves and fish [21,22,23]. Based on most recent toxicological studies, the European Commission deregulated PTXs in shellfish [24]. Therefore, the toxic potential and impact on the shellfish industry will be determined by the toxin profiles of the local strains [25,26]. The most damaging blooms for the shellfish industry are those of strains with a toxin profile containing only OA and DTXs, i.e., *D. acuminata* in Western Iberia. *Dinophysis* strains producing only PTXs are no longer a problem in countries where lipophilic toxins in shellfish are identified and quantified by analytical methods, and deregulation of PTXs has been enforced for national markets and exports.

Species/strains of *Dinophysis* differing in toxic potential and impacts adapt in distinct ways to changing environmental conditions [27]. Previous studies in fjords and upwelling systems have shown that *D. acuminata* and *D. acuta* have phenological differences [27,28,29,30,31], and are associated with different microplankton assemblages during annual succession [3]. When the two species do co-occur, their cell maxima are vertically segregated. All the above considerations point to the need to resolve uncertainties about the toxin profiles and content of local strains of *Dinophysis,* and to identify environmental conditions promoting bloom development of each species to support managerial performance. *Dinophysis* species–specific bloom patterns, and inter- and intraspecific variability in toxin profiles have been relatively unexplored in Chile with the exception of a few studies in northern Patagonian fjords [8,32,33].

Monthly hose sampling used in the regular monitoring programme does not provide detailed phenological observations or fine scale description of niche partitioning amongst *Dinophysis* species [34]. *D. acuta* is considered the main producer of diarrhetic shellfish toxins in Southern Chile.

This species has been associated with all major historic DSP events (reviewed in [35]), and with contamination with okadaic acid (OA) in addition to dinophysistoxin 1 (DTX1) and pectenotoxin 2 (PTX2). Different strains of *D. acuminata* have been reported in Chile with profiles containing only PTX2 between the Atacama (33.5° S) and Bío Bío regions (37° S) [36,37] or with two groups of lipophilic toxins (DTX1 in addition to PTX2) in the southernmost region, Magallanes [38]. Blooms of *D. acuminata* related to shellfish contamination with only PTX2 are also the most frequently reported in the Los Lagos region [39,40]. Recent establishment of *D. acuminata* cultures by Paredes-Mella et al. [41] with strains isolated from inland and oceanic waters of Los Lagos and Aysén, showed that these strains produced only PTX2, and cast doubt on the toxigenic nature of *D. acuminata* blooms in North Patagonian fjords.

The niche concept [42] is a useful approach to understand the relationship between a species distribution and its environment [43,44,45]. The niche of a given species has two components: the *fundamental niche*, i.e., the n-dimensional hypervolume within which the population of a species can persist, survive, and reproduce indefinitely without biological constraints, and the *realized niche*, or proportion of the fundamental niche within which the species actually persists. The realized niche takes into account the effect of abiotic variables and biological interactions [42]. The Outlying Mean Index (OMI) analysis proposed by Dolédec et al. [46] has proven to be a useful ordination method to describe phytoplankton species niches [47,48,49], including those responsible for HAB events [33,50,51]. This analysis, applied here to differentiate the niches of *D. acuminata* and *D. acuta*, had been used before when blooms of the two species coincided in Puyuhuapi Fjord in February 2019 [8]. 

During late summer 2020 (February–March), a bloom with overwhelming dominance of *D. acuminata* was observed in two NW Patagonian fjords, Puyuhuapi (PUY) and Pitipalena (PIT), differing in latitude, geomorphology, hydrodynamic conditions, and sampling season (Figure 1). Intensive 24 h sampling was carried out to: (i) estimate in situ division rates of *D. acuminata*; (ii) describe diurnal vertical migration (DVM) and/or identification of optimal layers for growth or aggregation of *Dinophysis* and its potential ciliate prey *Mesodinium*, and (iii) relate the distribution of lipophilic toxin profiles with *Dinophysis* species during the 24 h study. The ultimate goal was to compare environmental conditions favouring *D. acuminata* blooms in the summer of 2020 with those observed during an exceptional bloom of *D. acuta* in 2018 [32] and with the summer of 2019 when blooms of the two species co-occurred [8], or, in other words, to find out what were the environmental filters that promoted years with exceptional densities of one species and the exclusion of the other. These results contribute to the improvement of risk assessment of diarrhetic shellfish poisoning (DSP) events affecting shellfish in one of the world’s most productive aquaculture sites.

## 2. Results

### 2.1. Vertical Distribution of Thermo-Haline Properties

Oceanographic conditions in PUY and PIT fjords showed distinct stratification patterns (Figure 2). In PUY, strong salinity (S) gradients were formed in the top 5 m between a colder and fresher water layer (FW, S < 11) less than 2 m thick, and a warmer and saltier layer, the Estuarine Freshwater (EFW, S = 11–21) above the Estuarine Salty Water (ESW, S = 21–31) extending to ~20 m depth (Figure 2A,B). Density gradients and buoyancy frequency (Brunt–Väisälä frequency, 150 cycles h^−1^) were maximal in the top 5 m which depicted a thermal inversion all through the cycle study (Figure 2C).

PIT had a low freshwater inflow that year and showed no signs of thermal inversion associated with the FW layer. Instead, there was a warmer layer of EFW water at the surface (S < 2 m), and the ESW, down to 10–20 m, above the Modified Subantarctic water (MSAAW) (Figure 2D,E). Buoyancy frequency (~100 cycles h^−1^; Figure 2F) was lower than in the PUY fjord.

A tidal range of ~2 m with a semidiurnal pattern was observed in both fjords (Figure 3A,B). Water temperature at 2 m in PUY (14–15 °C) (Figure 3C) was slightly higher than in PIT (13–14.5 °C) (Figure 3D), a fjord where dissolved oxygen (DO) levels at the fixed station showed a large variability, with a maximum of 13 mg L^−1^ (Figure 3D). At a 6 m depth, temperature and DO values recorded in PIT were more homogeneous (Figure 3E,F), but values in PUY (~ +2 °C and +2 mg L^−1^) were higher. In addition, chlorophyll-*a* (Chl-*a*) showed a constant value of 10 μg L^−1^ in PUY, whereas in PIT, an extremely variable data set were recorded that ranged from 0 (under detection levels) to a maximum of 40 μg L^−1^) variable (Figure 3G,H).

Differences in the distribution of nutrients were also relevant (Figure 4). In PUY, nitrate and phosphate were depleted in the top 10 m. (Figure 4A). Nitrates in PIT were higher and harmonically distributed with a mean of 8.00 ± 4.17 µmol L^−1^ and a maximum of 13.84 µmol L^−1^ at 20 m (Figure 4E). Nitrites showed a different pattern, with higher values in PUY (0.70 ± 0.41 µmol L^−1^) and depleted during the day below 5 m (Figure 4B). Values in PIT were lower (0.41 ± 0.13 µmol L^−1^) and homogeneously distributed (Figure 4F). 

Phosphates, depleted at the surface and exhibiting a phospocline at 15 m depth, showed similar patterns to nitrates and a harmonic distribution in both fjords. Concentrations in PIT (1.13 ± 0.16 µmol L^−1^; max. 1.80 µmol L^−1^) were almost two-fold higher than in PUY (0.66 ± 0.61 µmol L^−1^) (Figure 4C,G). Silicates in PIT (8.69 ± 2.8 µmol L^−1^) (Figure 4H) were higher than in PUY (5.98 ± 4.2 µmol L^−1^) (Figure 4D), and there was an evident silicacline (7.5 µmol L^−1^ approx.) at 6 m.

### 2.2. Distribution of Dinophysis acuminata, D. acuta, Mesodinium, and Other Observations 

High cell densities (>3 × 10^3^ cells L^−1^) of *D. acuminata* were found in PUY within the warmer (14.5–15 °C) sub-surface (2–6 m) layer of Estuarine Salty Water (ESW; Figure 5A) and the cell maximum (3.8 × 10^3^ cells L^−1^), on February 19 at 08:00 h and 4 m depth, was at the interface between the EFW and the ESW water layers (Figure 5A). In PIT, *D. acuminata* cell maximum (1.5 × 10^3^ cells L^−1^) was found at 2 m depth within the surface layer (0–4 m), 13.4–14.0 °C on March 18 at 18:00 h, also in the interface between EFW (S = 11–21) and ESW (S = 21–31) (Figure 5D). Very low densities of *D. acuta* were observed in both fjords, with cell maxima not exceeding 2 × 10^2^ cells L^−1^ in PUY and under detection levels in PIT except during the first few hours of sampling (Figure 5B,E). *Mesodinium* species, putative prey of *Dinophysis*, were abundant in both fjords throughout the 24 h sampling. In PUY, *Mesodinium* spp. were found from 0 to 20 m at night and aggregated in the top 8 m during the day (max. of ~ 6.6 × 10^3^ cells L^−1^ at 16:00 h) (Figure 5C). In PIT, higher densities of *Mesodinium* (>10^4^ cells L^−1^) aggregated in the top (0–4 m) layer and the maximum was observed in the same time–depth window as in PUY (Figure 5F). 

Although not a target species in this study, high densities (>10^5^ cells L^−1^) of fish-killing dinoflagellates of the genus *Karenia* were observed in the samples from PIT which probably corresponded to the high Chl-*a* values observed at 6 m with the continuous sensor. Peaks and troughs observed in the circadian distribution of Chl-*a* and in DO values would be explained by the vertical migration of *Karenia* spp. and elevated DO values during the light hour and lower at night due to the balance between photosynthesis and respiration (Figure 3E,F).

### 2.3. Distribution of Other Potential Ciliate Prey

Species belonging to six genera of plastid-retaining ciliates (*Laboea*, *Leegaardiella, Lohmanniella*, *Paratontonia*, *Pseudotontonia,* and *Strombidium*), in addition to *Mesodinium*, were detected in both fjords (Figure 6). Differences between fjords were highest in the case of *Paratontonia* (max. 600 cells L^−1^ in PUY), distributed between 6 and 14 m (Figure 6D). *Strombidium*, a putative prey for *Dinophysis*, was observed in both fjords, with higher densities in PIT (max. 2.2 × 10^3^ cells L^−1^) after midday (Figure 6F). Some occasional overlapping of *D. acuminata* and potential alternative prey populations were observed. For example, in the PIT fjord, the time–depth distribution of *D. acuminata* (Figure 5D) coincided with that of *Laboea* spp., although maximal densities of the latter did not exceed 300 cells L^−1^ (Figure 6G).

### 2.4. Physiological Status of Dinophysis acuminata: Division Rates and Cellular Toxin Content

*Dinophysis acuminata* exhibited a phased-cell division in both fjords, cytokinesis occurring during a time-window from late dark to mid-day. Frequency of cells undergoing division (*f_c_*) and recently divided (*fr*) cells of *D. acuminata* were observed from 06:00 h to 12:00 h in PUY (*µ*= 0.29 d^−1^; µ_min_ = 0.19 d^−1^) and to 13:00 h in PIT (*µ*= 0.39 d^−1^; µ_min_ = 0.22 d^−1^), where division rates were higher (Figure 7). Maximal frequency of dividing cells (*fc*) was at 08:30 h in PUY (1.5 h after dawn) and at 09:00 (02:00 h after dawn) in PIT, and of recently divided cells (*fr*) at 10:00 and 10:30 respectively. Thus, the peak of division was in both cases (1.5–2 h) after dawn, and division time (T_D_) lasted 1.5 h. 

Liquid chromatography Coupled to High Resolution Mass Spectrometry (LC-HRMS) analysis of net tow extracts showed that DTX1 and PTX2 were present in most samples containing *D. acuminata* in both fjords. OA was not detected. Analyses revealed a chromatographic peak with a retention time of 4.79 min with an [M-H]^-^ ion 817.4762 *m/z*. The fragmentation spectrum showed the major product ion 255.1236 *m/z* that confirms the presence of DTX1 (Figure A1). DTX1 was detected in 14.1% of the samples from PUY (maximum of 2.405 ng DTX1 NT^−1^ (net tows) on February 19, at 13:00 h) and in 34.6% of those from PIT (maximum of 1.718 ng DTX1 NT^−1^ on March 18, at 18:00 h) (Figure 8). The second chromatographic peak detected at 7.48 min with a parent mass [M+NH_4_]^+^ 876.5090 *m/z* corresponded to PTX2. The characteristic MS/MS fragment confirmed the presence of this toxin at m/z 841.4730, 823.4620, 805.4518, and 787.4406 *m/z* (Figure A1). PTX2 was detected in 75.6% of water samples from the PUY fjord with a maximum of 80.480 ng PTX2 NT^−1^ on February 19, at 18.00 (Figure 8), and in 87.2% of those from PIT where the maximum detected was 42.161 ng PTX2 NT^−1^ on March 19, at 07:00 (Figure 8).

### 2.5. Niche Analysis

The Outlying Mean Index analysis (OMI) revealed that the two *Dinophysis* species, *D. acuminata* and *D. acuta*, had a significant realized niche (*p* < 0.01) in PUY and PIT fjords. Regarding the potential prey community, four out of seven ciliate species had significant realized niches in PUY and five out of seven in PIT (Table 1). Notwithstanding, some differences were found in the realized niches of *Dinophysis* on each fjord. 

PUY fjord: The first and second axis of the OMI ordination (PC1 and PC2) encompassed 96.9% (OMI1: 73.02% and OMI2: 23.94%) of the total projected inertia (Figure 9A). The OMI plane allowed us to identify one diagonal gradient from the top right (characterised by the Brunt–Väisälä frequency–Depth–Salinity) of brackish estuarine and stratified surface water to the bottom-left of deeper, saltier, and less stratified water (Figure 9B). The second gradient was only represented by the temperature. The niche of *D. acuminata* and *D. acuta* was defined by warm sub-surface and estuarine (saltier than the FW) water, with differences between species related to the first (OMI1) axis. *D. acuminata* cells spread between isohalines 21 and 27 and *D. acuta* between 25 and 30. Both species were associated with a high water-column stratification, but not with the layer of maximal, salinity-driven, density gradients at the surface.

The OMI parameters (Table 1) showed a marginality value for *D. acuminata* (OMI = 0.30) three times lower than *D. acuta* (OMI = 0.97), indicating a more suitable environment for *D. acuminata* during the sampling period. Furthermore, *D. acuminata* showed a higher niche breadth (Tol = 1.15) than *D. acuta* (Tol = 0.76), which explains the broader distribution of *D. acuminata* in space and time as compared to *D. acuta* (Table 1, Figure 9C). *Mesodinium* (putative *Dinophysis* prey) had the lowest marginality (OMI = 0.07) and the broadest niche breadth (Tol = 2.68), indicating its wider distribution range throughout the environmental conditions in which *Dinophysis* species were found. Results from the PERMANOVA analysis showed that temperature significantly (*p* < 0.05) explained the variability in distribution of *Dinophysis* species (Table 2).

PIT fjord: OMI analysis from this fjord showed that the first and second axis of the ordination (PC1 and PC2) contained 93.2% (OMI1: 85.68% and OMI2: 7.58%) of the total projected inertia (Figure 9D). The OMI plane was defined by two diagonal gradients, each one with inversely correlated variables (Figure 9E). The first gradient (composed by Temperature–Salinity) was of warmer and fresher estuarine water from the top left to colder and saltier water on the bottom right of the plane. The second gradient (Depth–Brunt–Väisälä frequency) was of deeper and less stratified water from the top-right to shallow and highly stratified water in the bottom-left.

Most species (i.e., *D. acuminata* and potential ciliate prey) were distributed along the Depth–Brunt–Väisälä gradient (Figure 9E). The niches of *Dinophysis* and ciliates species were characterised by upper layers of the water column, with high stratification, and where marked salinity gradients occurred. The niche of *D. acuminata* was mainly defined by surface waters, with warmer temperatures and low salinity compared to deeper waters, and an important influence of the high stratification that characterise the interface between the EFW and the ESW within the first 5 m. The niche of *D. acuta* was predominantly defined by surface waters, warm temperatures, and estuarine salty water (ESW, S >21), but by higher salinity and depth than *D. acuminata*.

The OMI parameters showed a marginality for *D. acuminata* (OMI = 2.43) less than half the value found for *D. acuta* (OMI = 5.59), indicating the most favourable conditions for *D. acuminata* during the sampling period. In addition, the latter showed a higher tolerance value (Tol = 2.78) than *D. acuta* (Tol = 1.23). Tolerance values indicate a broader realized niche breadth for *D. acuminata* and a narrower one for *D. acuta* (Table 2, Figure 9F). The ciliate *Mesodinium* had a very low marginality (OMI = 0.07) and a very high niche breadth (Tol = 3.83), suggesting that it could be present through the habitat occupied by the two *Dinophysis* species. Finally, PERMANOVA analysis showed that temperature and salinity were the significant variables (*p* < 0.05) that contributed to most of the explained variability of *Dinophysis* species (Table 2).

## 3. Discussion

Blooms of the *Dinophysis acuminata* followed by blooms of *D. acuta* are common in coastal temperate seas. Some of the regions most affected are the European Atlantic Arc [25,53], the Patagonian fjords, Southern Chile [35], and the sounds and fjords in New Zealand South Island [54]. 

The comparative bloom dynamics of *D. acuminata* and *D. acuta* have been addressed in Swedish [30,55], Scottish [30,55], and Western Ireland [56] fjords and embayments [30,55] and in the Galician Rías and Portuguese coastal lagoons, Western Iberia [27,28,29,57,58]. In these regions, endemic populations of *D. acuminata* and *D. acuta* with differing phenologies and toxic potential contaminate shellfish with diarrhetic toxins and pectenotoxins, and cause lengthy harvesting bans when their concentrations in shellfish exceed regulatory levels. Bloom dynamics are site specific [3], and toxic potential varies according to the toxin profiles and contents of local strains of *Dinophysis,* and according to the uptake and depuration kinetics of the affected shellfish resources [25,26].

### 3.1. Oceanographic Settings That Promote D. acuminata/D. acuta Dominance 

Habitat conditions of *D. acuminata* and *D. acuta* have been relatively unexplored in the Chilean fjords. Here, we compare microstructure condition in summer 2020 in northern Patagonia with those that led to: (i) the dominance of *D. acuta* in summer 2018 when this species reached record densities (660 × 10^3^ cells L^−1^) [32]; (ii) the co-occurrence of *D. acuminata* and *D. acuta* in summer 2019 [8] and (iii) the dominance of *D. acuminata* in both fjords during summer 2020 (*D. acuta* failure) (Figure 10).

During 2018, there were positive anomalies in the summer air (+1 °C) and sea surface temperature (SST) (+2 °C), and drought following increased precipitation in late spring. This situation led to a very warm (16–18 °C) top layer (0 to 5–7 m) of EFW delimited by the 21 isohaline, and a record *D. acuta* bloom in PUY [32]. *D. acuta* maxima were in deeper and cooler (15–16 °C) more saline ESW with S > 25 (Figure 10 A,B).

In summer 2019, SST in the top 4–6 m was slightly above 16 °C and salinities of 17–20 were similar to 2018, i.e., within the upper limits of the EFW range (S = 11–21). Dense populations of *D. acuta* and *D. acuminata* co-occurred in the same fjord; *D. acuminata* maxima were above the pycnocline (interface between EFW and ESW), and those of *D. acuta* were deeper, within the pycnocline [8].

Oceanographic conditions in PUY during summer 2020 (this work) included cold (12.5 °C) FW (S < 11) between 0 and 2 m depth forming strong haline gradients with a thermal inversion (SST, 2 °C colder at the surface than at 5 m depth) (Figure 2A,B). 

Comparison of summer conditions in PUY for the two extreme years, 2018 with only *D. acuta* bloom and 2020 with only *D. acuminata,* suggests that i) the presence of cold FW and thermal inversion were the key hydrographic features preventing *D. acuta* growth in 2020, and ii) SST over 18 °C may be excessive for *D. acuminata,* which is then replaced by *D. acuta*.

The *Dinophysis* growth season at different sites on Atlantic European coasts has been related to thermal stratification. Blooms of *D. acuta* in SW Ireland [56] and Galicia [17], initiated in mid-shelf waters, share common physical environment requirements: persistent thermal stratification (~2 month) and water column stability met in late summer. Cell maxima and thin layers are formed in the pycnocline region above the *chl a*-maximum. Analysis of a 30 y time series of *D. acuta* monitoring counts and environmental condition in a seasonal-upwelling area showed no specific trends but a relationship between exceptional events and large-scale climatic anomalies reflected in a latitudinal displacement of the Azores High and the Icelandic Low pressure centres [58].

Thermo-haline layering in fjords is largely determined by the balance between solar heating and precipitation (rainfall), ice-melt, and rivers’ streamflow [52,59]. There is only one field study before 2018 providing circadian and seasonal distribution of *Dinophysis* with depth in Pitipalena [60]. In this study, *D. acuminata* was observed in spring–summer (October to February) in a wide range of temperature (10.5–19.60 °C) and salinity (4.75–30.73)conditions, but peak concentrations were recorded in narrower ranges (T: 11.2–12.5 °C; S: 19.8–22.6). *D. acuta* was found only once, in a surface sample collected near the head of the fjord in summer 2005 (T: 19 °C; S: 15.2) associated with shear instabilities induced by tidal forcing.

On a different spatio-temporal scale, analysis of an 11-year (2006–2017) time series of monthly monitoring (0–10 m and 10–20 m hose samples) data from Reloncaví Fjord (REL), north of PIT, Los Lagos region, by Alves de Souza et al. [33], revealed that *D. acuminata* blooms, presumably advected with oceanic water into the fjord, were favoured by cold years (La Niña) with lower river outflow (SAM positive). Baldrich et al. [34] analysed the interannual variability of *D. acuminata* and *D. acuta* in REL, PUY, and PIT between 2006 and 2018, and found a common decreasing trend in rainfall and river outflow, but site-specific differences in stratification patterns. *Dinophysis* responses to these differences were species- and site-specific. *D. acuminata* bloomed in the whole region with increasing intensity in the southernmost fjords (Magallanes). There was an apparent increase in *D. acuta* bloom intensity from 2011 to date in PUY. This fjord, with the lowest rainfall and river discharge, highest water residence time, and the only one with no FW layer (S < 11) in summer, was apparently the only one with suitable conditions for *D. acuta* blooms.

The two case studies above show the importance of species- and site-specific differences and the need to consider the balance between multiple scale phenomena (SAM, ENSO, local climate) acting during the right seasonal window for the target species (spring for *D. acuminata*, summer for *D. acuta*). Results here illustrate how these different scenarios led to important differences in vertical gradients. However, all studies point to the unsuitability for *D. acuta* blooming of PIT and REL, and the negative effect of the FW layer.

Fjords have a marked salinity-driven stratification all the time; *D. acuminata* growth is not triggered until temperature increases and the thermally-inverted FW layer thins [8,33,60]. As in other estuarine systems, *D. acuta* in fjords is associated with warmer and dryer weather and with thermal stratification. In the scenarios reviewed, the onset of *D. acuta* growth requires the disappearance of the FW layer and the thermal inversion, and the establishment of a warm surface layer above the thermocline. These features could provide short-term forecasts of no risk of *D. acuta* bloom. Likewise, increasing SST and thinning of the FW would announce risk of *D. acuminata* bloom initiation, provided an inoculum and prey were available [33].

Nitrate and phosphate concentrations, nearly exhausted in the upper layer, were higher in the subsurface layer in PIT than in PUY. However, *Dinophysis acuminata* and *D*. *acuta* prefer ammonia, urea, and other regenerated N sources and do not take up nitrates [61,62]. Both *Dinophysis* and ciliate prey *Mesodinium* are obligate mixotrophs with high prey selectivity [63]. Prey availability is the main biotic factor promoting growth, and often the main constraint in field populations [64]. Results from laboratory cultures suggest that nitrate and phosphate can be used by *Mesodinium rubrum* for enhanced growth provided that plastids are obtained from their prey [65]. It is clear that in 2020, *Mesodinium* and other potential ciliate prey were present in adequate concentrations and were not limiting *Dinophysis* growth.

### 3.2. Toxin Diversity

Lipophilic toxins PTX-2 and DTX-1 were present during the summer 2020 sampling in PUY and PIT fjords when *D. acuminata* was the overwhelmingly dominant species of *Dinophysis.* OA was not detected; its absence in net tow extracts coincided with extremely low cell densities of *D. acuta* in PUY (<200 cells L^−1^), and its absence (except at the beginning of the study) in PIT. 

A contrasting situation was that in summer 2018, when a record bloom of *D. acuta* reached densities over 6.6 × 10^5^ cells L^−1^ in a thin layer extending over 15 km in PUY, *D. acuta* represented between 96 and 99% of the total *Dinophysis* counts, and OA (dominant), DTX1 and PTX2 were present in all samples. An intermediate situation was found in summer 2019, when blooms of *D. acuminata* and *D. acuta* co-occurred but their populations were vertically segregated. Analyses of toxins in bottle samples at specific depths showed OA, DTX1, and PTX2 in samples collected at the depth of the *D. acuta* maximum and only PTX2 in samples from the *D. acuminata* maximum These results support previous studies pointing to *D. acuta* as the only putative OA producer in the Patagonian fjords [8,32]. It also confirms that although PTX2 was the only major toxin associated with dense blooms of *D. acuminata* in different Chilean region, DTX1 can be present in other Patagonian strains of this species. Earlier studies reported the presence of DTX1 associated with *D. acuminata* blooms in the regions of Los Lagos and Magallanes [38,39,66]. Likewise, Suárez-Isla et al. [31] reported the occasional presence of DTX1 associated with *D. acuminata* in the intensive toxin monitoring of mussel exports carried out around Chiloé Island (Los Lagos). More recently, Paredes-Mella et al. [41] reported PTX2 as the only toxin present in cultures of six strains of *D. acuminata.* Of these, three were isolated from oceanic Pacific waters, south of Chiloé (Los Lagos), and the other three in Moraleda Channel off Pitipalena Fjord (Aysén). Images of the strains showed specimens with round antapical ends, quite similar to the *D. ovum*-like specimens from Bío Bío, Central Chile associated with PTX2 contamination in shellfish in that region (see Figure 2B in Díaz et al. [36]). 

Whether DTX1 is produced by different strains of *D. acuminata* (genetic variability) or by the same strains with a different expression under changing environmental conditions should be clarified. *D. acuminata* strains associated with PTX2 contamination of shellfish are frequent along the entire Chilean coast where OA has been related to the occurrence of *D. acuta*. This trend is confirmed here: OA was not detected, and *D. acuta* was absent or just above detection levels (100 cells L^−1^) in the two fjords visited. Results here provide new evidence of the coexistence of *D. acuminata* strains with different toxin profiles in northern Chilean Patagonia (Los Lagos and Aysén regions), from 41.5 to 55.9° S (reviewed in [35]).

### 3.3. Niche Analysis

The OMI analysis showed that the environmental variables included (i.e., depth, temperature, salinity, and Brunt–Väisälä frequency) defined the niches of the two main *Dinophysis* species in the two fjords. The OMI marginality value for *D. acuta* was three times higher than for *D. acuminata* in PUY and its tolerance 40% lower; in PIT, marginality of *D. acuta* doubled that of *D. acuminata* and its tolerance was less than half (Table 1). These OMI values revealed that environmental conditions in both fjords, in particular in PUY, were favourable for *D. acuminata*. Thus, the environmental conditions found in the cycle study represented a typical habitat of *D. acuminata* leading to its dominance in PUY and PIT fjords. The suitable conditions were confirmed by the moderate to good division rates (*µ*= 0.3 and 0.4 d^−1^ in PUY and PIT, respectively) of *D. acuminata* in the two fjords. However, in comparison with *Mesodinium*, the “sentinel” species occupying practically the full realized niche in the OMI analysis (Tol = 2.68) in the 2020 data set from PUY, *D. acuminata* tolerance (Tol = 1.15) suggests its habitat had not reached its optimal conditions. 

*D. acuta*, with the highest marginality values within the full list of organisms, seemed to be in an unfavourable habitat in PUY 2020. During the 2019 blooms, *D. acuta* doubled more than once per day (*µ*= 0.76 d^−1^), and tolerance values were 3.11 for *Mesodinium*, 1.90 for *D. acuminata,* and 1.49 for *D. acuta*, i.e., the tolerance values for the two *Dinophysis* species in 2019 were almost double those observed in 2020.

OMI and PERMANOVA analysis supported the influence of temperature and salinity on the distributions and cell densities of *D. acuminata* and *D. acuta.* The niche of *D. acuminata* showed that warm temperatures in the broader depth interval in the PUY fjord (13 °C isotherm at 12 m), where sampling was carried out earlier in late summer, favoured higher cell densities compared to those recorded in PIT, where the same isotherm was at 5 m. *D. acuminata* was restricted to the upper end of the ESW layer, its cell maximum in the interface between ESW and EFW (S < 21). *D. acuta* was within the same ESW, but in deeper layers with higher salinity (S > 25), far from the interface with EFW. There were no cells in the estuarine freshwater (EFW, S = 11–21), so that *D. acuta* seemed to avoid salinities < 21 in the two fjords. 

*Mesodinium* species are the only confirmed prey of *Dinophysis* and the source of their kleptoplastids in the field and laboratory cultures [67]. The OMI parameter values from the two fjords showed that environmental conditions during the 24 h studies were suitable and common to *Mesodinium* species and *D. acuminata.* The two species shared a broad niche breadth (Figure 5C,F). These observations are common with those from earlier studies in the Galician Rías [7,68] and the Baltic Sea [69], showing that *D. acuminata* and *Mesodinium* have their own specific niches, and that while the dinoflagellate may aggregate in a thin layer at the surface, it eventually intercepts the ciliate *Mesodinium* when the latter performs its diurnal vertical migration. 

Results here support the view of *D. acuminata* as a cosmopolitan species adapted to tolerate a wide range of temperature and salinity conditions [70,71] (Figure 9C,F). The niche of both species was strongly influenced by temperature, pointing to the importance of thermal stratification, in particular for the presence of *D. acuta*. However, the two species had their maxima in restricted conditions of T and S, most likely derived from their behavioural traits. These include aggregation in thin layers that favour the sexual encounter of gametes, interception of ciliate prey, and secretion of allelopathic substances to reach a threshold concentration [70,71]. 

At this point, it is important to be aware of the characteristics of the realized niche of the two *Dinophysis* species under examination. *D. acuminata* presents a high tolerance and broader niche breadth than *D. acuta* in most scenarios, but both species usually have a high marginality when compared with the accompanying species of the assemblage. In other words, *D. acuminata* is able to persist in wide ranges of temperature, salinity, and light intensity, but requires a precise set of these properties for optimal growth. These properties can be identified by examining vertical profiles of the species and the precise water layer where cell maxima are located and good division rates observed. That being said, and revising the location of rapidly dividing *D. acuminata* populations in the fjords, conditions were surprisingly constant. The optimal window for *D. acuminata* was at the interface between the EFW and ESW layers, i.e., salinity ~ 20 ± 2 and temperature of 13 ± 2 °C. The optimal window for *D. acuta* was always observed in deeper and saltier waters but once the FW had vanished and stronger thermal stratification developed. 

Data providing wide ranges of temperature and salinity where *D. acuminata* is observed in a given locality are not meaningful if they refer to net tows or tube samples. These samplers may go through marked density gradients and ignore the precise conditions where the cells are actively growing or just “tolerating” prey deprivation, overwintering, or other situations.

## 4. Materials and Methods

### 4.1. Study Area

The southern coast of Chile, also known as the Chilean Patagonia (41 to 55° S), constitutes one of the most extensive fjord and channel systems in the world (Figure 1). This system has a rugged bathymetry, dissected coastline, and strong but highly variable water column stratification. Marked salinity gradients exhibit seasonal and latitudinal patterns determined by heavy riverine inflow from ice melting in late spring (November–December) and persistent rainfall with an average of 2700 mm y^−1^ and up to 5000 mm in exceptional years [72,73]. Summer heating breaks the winter–spring thermal inversion and generates stronger thermohaline gradients [59].

Circulation in the Patagonian fjords is of two-layered estuarine-type, with a variable (5–10 m) estuarine surface water (EW) and a more uniform saltier lower layer, the Subantarctic Water (SAAW, S > 33) reaching 150 m depth [74]. Mixing of the two layers at the interface generates Modified Subantarctic Water (MSAAW, S = 31–33) [59]. Depending on freshwater inputs, different water masses can be identified within the estuarine surface water: Estuarine Fresh Water (EFW, S = 11–21), Estuarine Saline Water (ESW, S = 21–31). When salinity is less than 11, the water is classified as Fresh Water (FW) [52]. The Pitipalena (PIT) (~ 43°S), and Puyuhuapi (PUY) (~ 44° S) fjords, in the Aysén region, form part of this great fjord system. PUY (100 km) is much longer than PIT (22 km), the latter being more semi-enclosed [59,60,75]. Unlike PIT fjord, PUY has two connections with oceanic waters, one through the Moraleda Channel in the mouth and another through the Jacaf Channel close to the head [59]. In the two fjords, the main freshwater inputs come from riverine inflows and rainfall. The main river flowing into PUY (Cisnes, average river discharge 218 m^3^s^−1^) has its mouth located by the middle reaches of this fjord [75,76]. In contrast, in PIT, the Palena river, with a four-fold average river discharge (800 m^3^s^−1^), is located at the mouth [60]. These characteristics affect hydrodynamic conditions, including stratification and water residence time, which is maximal in PUY (~250 days) compared to PIT fjord (~200 days) [77,78], directly promoting phytoplankton retention and HAB development.

### 4.2. Field Sampling

Measurements of abiotic and biotic factors were carried out hourly or every 2 h during intensive 24 h surveys at a fixed station on each fjord. The aim was to study small-scale interactions that modulate the time–depth distribution of *Dinophysis* and potential ciliate prey populations in two NW Patagonian fjords. The circadian variability, potential vertical migration, and specific division rate of *Dinophysis* species were evaluated in PIT (March 18–19) (Figure 1B) and PUY (February 18–19) (Figure 1C) fjords, two coastal sites with distinct hydrodynamic conditions, but subject to recurring toxic outbreaks. Sampling in 2020 was carried out in summer, i.e., the season with the highest probability of occurrence of these events [34]. In both surveys, observations on division phases began at 18:00 h.

Vertical profiles of temperature, salinity, dissolved oxygen, and in vivo chla fluorescence were obtained with an RBR CTD (conductivity–temperature–depth) profiler (https://rbr-global.com) model Concerto3 equipped with a Turner Designs CYCLOPS-7 fluorometer (excitation 460 nm, emission 620–715 nm). The (CTD) probe was cast hourly to 50 m depth. CTD data processing was carried out with the software provided by the manufacturer and depicted using the Ocean Data View software version 5.1 [79]. Additionally, temperature, dissolved oxygen, and chlorophyll-*a* sensors were installed at 2 m and 6 m at the fixed sampling locations in PUY and PIT fjords. A MiniDOT oxygen logger (www.pme.com) was used to obtain the temperature and dissolved oxygen records. In addition, a C-Fluor logger model CYCLOPS-7 (www.pme.com/products/cfluor-logger, accessed on 9 December 2022) was installed at 6 m.

Water samples of 100 mL for quantitative analysis of *Dinophysis* species and their potential micro-ciliate prey were collected every 2 h at 2 m intervals, from the surface to 20 m, and at 25 m, and 30 m, with a 5 L Niskin bottle, and immediately fixed with neutral Lugol’s iodine solution [80].

Plankton nets (20 μm mesh) were towed vertically from 20 m to the surface twice every hour. Samples of 100 mL were taken from the first tow and fixed with neutral Lugol’s solution [80]. These samples were used to estimate frequency of cells undergoing mitosis and in situ daily specific division rates (*μ*) (see Section 4.5). A second net tow was collected for toxin analyses. The whole content of the net collector was filtered through Whatman GF/F fiberglass filters (25 mm Ø, 0.7 μm pore size) (Whatman, Maidstone, England), the filters and filtered material placed in a cryotube, mixed with 1 mL analysis grade methanol, and stored in the laboratory at −20 °C until analysis.

### 4.3. Nutrients

Water samples for nutrient analysis (NO_3_^−^, NO_2_^−^, PO_4_^3−^, and Si (OH)_4_) were taken every 6 h using 50 mL syringes directly connected to the spigot of the Niskin bottle at each sampling depth (i.e., 0, 4, 8, 12, 16, and 20 m). Samples were kept in a cooler on board and taken to the local laboratories (maximum of 2 h between collection and filtering), filtered through Whatman GF/F fiberglass filters (47 mm Ø, 0.7-μm nominal pore size) (Whatman, Maidstone, England), and the filtered material frozen at −20 °C until analysis. Dissolved inorganic nutrients were analysed using a Seal AA3 AutoAnalyzer, according to Grasshoff, et al. [81], and seawater analysis according to the standard method of [82]. Ammonia analyses were omitted for logistic reasons, i.e., the impossibility to ensure analyses of this labile molecule very soon after collection in remote areas in southern Chile.

### 4.4. Phytoplankton Analysis

For quantitative analyses of phytoplankton, 10 mL of each Lugol-fixed bottle sample were left to sediment over 6 h. Samples were analysed with an inverted microscope Olympus CX40 (Olympus, Japan), according to Utermöhl [83]. To count *Dinophysis* species and ciliates, the whole surface of the chamber was scanned at a magnification of 100 X so that the detection limit was 100 cells L^−1^. Ciliates were identified to genus or species level when possible using magnifications of 200× and 400×.

### 4.5. Division Rates

Aliquots (1 mL) of each Lugol-fixed net, collected every 1 h, were used to estimate the in situ division rates, with a “post-mitotic index approach”, from the frequency of dividing (paired) and recently divided (incomplete development of the left sulcal list) cells, which were recognised by their distinct morphology (see Figures 2 and 14 in Reguera et al. [84]) following the model of Carpenter and Chang [85]:(1)$$\mu =\frac{1}{n\left(T_{c}+T_{r}\right)}\overset{n}{\underset{i=1}{\sum }}{\left(t_{s}\right)}_{i}\ln\left[1+f_{c}\left(t_{i}\right)+f_{r}\left(t_{i}\right)\right]$$
where *µ* is the daily average specific division rate; *fc(t_i_)* is the frequency of cells in the cytokinetic (or paired cells) phase (*c*); and *fr(ti)* is the half frequency of cells in the recently divided (incomplete development of the left sulcal list) (*r*) phase in the *i^th^* sample. *Tc* and *Tr* are the duration of the *c* and *r* phases, considered as “terminal events” sensu [85] in this work; *n* is the number of samples taken in a 24 h cycle; and *ts* is the sampling interval in hours. The duration of the selected terminal events, *Tc + Tr,* was estimated as the interval of time necessary for a cohort of cells to pass from one phase to the next; in this case, the time interval between the time *t_0_*—when the frequency of cells undergoing cytokinesis, ƒc, is maximum—and the time *t_1_* when the fraction of recently divided cells *ƒr* is maximum:(2)$$\frac{1}{2}\left(T_{c}+T_{r}\right)=\left(t_{0}-t_{1}\right)$$
where *Tc*, *Tr*, *t_1_*, and *t_0_* are calculated after fitting a 5th-degree Gaussian function to the frequency data.

### 4.6. Lipophilic Toxin Analysis

#### 4.6.1. Toxins Sample Extraction

Tubes with the filtered net tow samples with methanol were centrifuged (20,000 g; 20 min), and sonicated with an ultrasonic cell disruptor Branson Sonic Power 250 (Danbury, CT, USA). The extract obtained was clarified by centrifugation (20,000 g; 15 min) and filtered through 0.20 μm Clarinert nylon syringe filters (13 mm diameter) (Bonna-Agela technologies, Torrance, CA, USA). To analyse free Okadaic Acid (OA) and other lipophilic toxins, an aliquot of 0.5 mL of each sample extract was placed in an amber vial and stored at −20 °C until analysis. To detect esterified OA-group toxins, 0.5 mL aliquots were subjected to alkaline hydrolysis following the standard procedure of the EU Reference Laboratory for Marine Biotoxins (EURLMB, 2015). Finally, samples were placed in an amber vial and stored at −20 °C until analysis.

#### 4.6.2. Toxin Detection and Quantification

Detection of lipophilic toxins in the extracts was carried out by LC-HRMS analyses following a modification of the method described by Regueiro et al. [86]. This modification allows us to use a shorter column and gives enough time for the elution of all lipophilic toxins. The instrumental analysis was developed using a Dionex Ultimate 3000 UHPLC system (Thermo Fisher Scientific, Sunnyvale, CA, USA). A reversed-phase HPLC column Gemini NX-C18 (50 mm × 2 mm; 3 μm) with an Ultra Guard column C18, both from Phenomenex (Torrance, CA, USA), was used. The flow rate was set to 0.35 mL min^−1^ and the injection volume was 10 μL. The mobile phase was used in gradient mode as follows: 85% of eluent A (100% water containing 6.7 mM NH_4_OH) and 15% of eluent B (90% acetonitrile:10% water with 6.7 mM NH_4_OH) was held for 1 min, followed by a linear increase to 80%B for 2.85 min, and then an increase to 85%B for 0.15 min, 90% B for 0.75 min, and 100% B for 3.25 min. Finally, the gradient returned to initial conditions over 2 min, and the column was re-equilibrated for 1 min. 

The detection of lipophilic toxins was carried out by a high-resolution mass spectrometer Q Exactive Focus equipped with an electrospray interphase HESI II (Thermo Fisher Scientific, Sunnyvale, CA, USA). The HESI was operated in negative ionisation mode with a spray voltage of 3 kV and in positive ionisation mode with a spray voltage of 3.5 kV. The temperature of the ion transfer tube and the HESI vaporiser were set at 200 and 350 °C, respectively. Nitrogen (>99.98%) was employed as sheath gas and auxiliary gas at pressures of 30 and 4 arbitrary units, respectively. The data was acquired in Selected Ion Monitoring (SIM) and data dependent (ddMS^2^) acquisition mode. All the analyses were performed with mass inclusion list, including the precursor ion masses, expected retention time window, and collision energy (CE) for each toxin (Table 3). In SIM mode, the mass scan range was set at *m/z* 100–1000 with a mass resolution of 70,000, the automatic gain control (AGC) was established at 5 × 10^4^, and the maximum injection time (IT) was 3000 ms. For dds^2^ the mass resolution was set at 70,000, AGC at 5 × 10^4^, and IT 3000 ms. In both cases, the isolation windows were 2 *m/z*. The toxin concentration in the extracts was quantified by comparing the area or the peaks obtained in the chromatograms with those of certified reference materials obtained from the NCR, Canada. 

Calibration curves for the three major toxins expected, OA, DTX1, and PTX2, were generated by plotting the area of the toxin standard peak versus its concentration in ng mL^−1^ (Figure A2). Limit of detection (LOD) and quantification (LOQ) for each toxin were estimated according to the equations
*LOQ= (blK* ± 10 × SD)/*m*
*LOD*= (*blK* ± 3 × SD)/*m*
where *blk*= blank signal; SD= blank average standard deviation; *m* = slope of the calibration curves.

### 4.7. Niche Analysis

Biological variables (i.e., *Dinophysis* and their potential plastid-retaining ciliate prey) were log-transformed (ln(x + 1)), and abiotic environmental variables (i.e., water temperature, salinity, depth, and Brunt–Väisälä frequency) were standardized as in Baldrich et al. [8]. An Outlying Mean Index (OMI) analysis was used to identify the ecological niche of *D. acuminata* and *D. acuta* [46]. The OMI analysis gives information about the realized niche and niche breadth of a given species in an n-dimensional space through OMI, Tol, Inertia, and RTol parameters. The statistical significance of the OMI analysis was tested using Monte Carlo permutations (10,000 permutations) [87]. Details of each OMI parameter are described in Baldrich et al. [8].

Finally, a marginal Permutational Analysis of Variance (PERMANOVA), based on Euclidean distances [88] was performed to identify the influence of environmental variables on *D. acuminata* and *D. acuta* following what is described in Baldrich et al. [8].

Statistical analyses were made using the ‘niche’ function in the ade4 package [89], and ‘adonis2′ function of the vegan package [90] of the free CRAN repository from the statistical and programming R software [91]. All data were graphed using the Ocean Data View software [79] and the statistical and programming R software [91].

## 5. Conclusions

High-resolution sampling has allowed us to identify some habitat preferences of two HAB species in stratified fjords. Interannual variations in summer stratification patterns act as a physical filter for motile *Dinophysis acuminata* and *D. acuta.* Comparison of vertical thermohaline microstructure in a year with strong predominance of *D. acuminata* (this work) with a year with exceptional blooms of *D. acuta* and years with co-occurring blooms of the two species allowed us to describe the optimal depths at which cell maxima are found, and sheds light on specific adaptations.

*Dinophysis acuminata,* predominant in Puyuhuapi and Pitipalena fjords in 2020, exhibited its maxima at the interface between estuarine freshwater (FW, S = 11–21) and estuarine salty water (ESW, S = 21–31). This species is found in a wide range of temperature and salinity. Nevertheless, the optimal depth for *D. acuminata* has been confirmed in different fjords, seasons, and scenarios in northern Patagonia. Niche analysis by the outlying mean index (OMI) method explains this by showing this species has a high tolerance but its realized niche requires a much narrower range of conditions (high marginality). 

*D. acuta* bloom development appeared constrained by thermal inversions, and the presence of surface freshwater (FW, S < 11). These physical conditions may be used to provide short-term forecasts of no risk of *D. acuta* bloom. The same species of *Dinophysis* have strains adapted to distinct sets of temperature and salinity in fjords, upwelling regions, and coastal lagoons in temperate seas. But in all situations, comparison of *D. acuminata* with *D. acuta* give similar results: *D. acuminata* is adapted to grow closer to the surface, in more brackish better lit conditions during spring–summer, as soon as a shallow thermohaline stratification develops; *D. acuta* is a late summer species, its maxima always in deeper waters with well-established thermoclines and higher buoyancy frequency.

The lack of detection of OA in two different fjords during blooms of *D. acuminata,* and the detection levels of *D. acuta* support the view that *D. acuta* is, so far, the only species of *Dinophysis* associated with okadaic acid in Chile. It also confirms that blooms of *D. acuminata* containing DTX1 (in addition to PTX2) coexist in Northern Patagonia with harmless blooms containing only PTX2. The spatio-temporal distribution of toxic (DTX1 content) strains of *D. acuminata* throughout Chilean Patagonia should be the topic of future studies.

## Figures and Tables

**Figure 1 marinedrugs-21-00064-f001:**
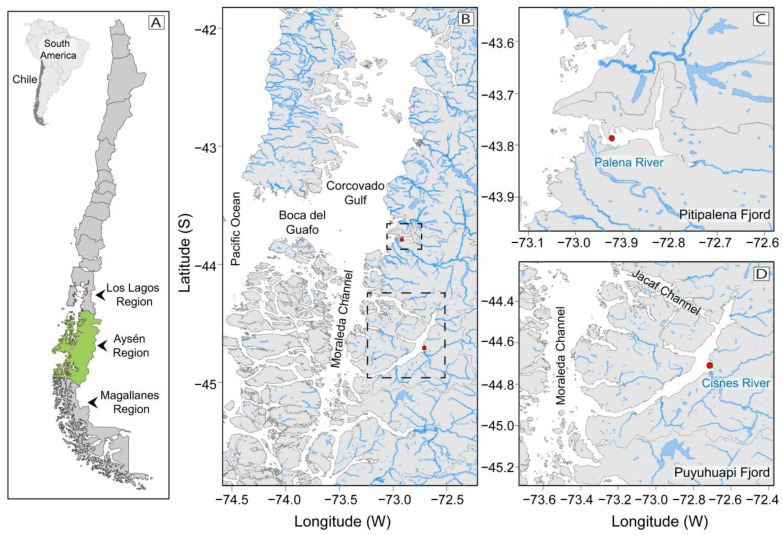
Map of the study area: Chilean coastline, and the three Patagonian provinces of Los Lagos, Aysén (shown in green), and Magallanes (**A**); Northern Patagonia channels and fjords system and their connections to Pacific Ocean. Pitipalena (PIT) and Puyuhuapi (PUY) fjords are framed in broken lines (**B**); Location of fixed sampling stations (red dots) for the 24 h studies in PIT (**C**), and in PUY (**D**) fjords, and their main freshwater sources (Palena and Cisnes River, respectively).

**Figure 2 marinedrugs-21-00064-f002:**
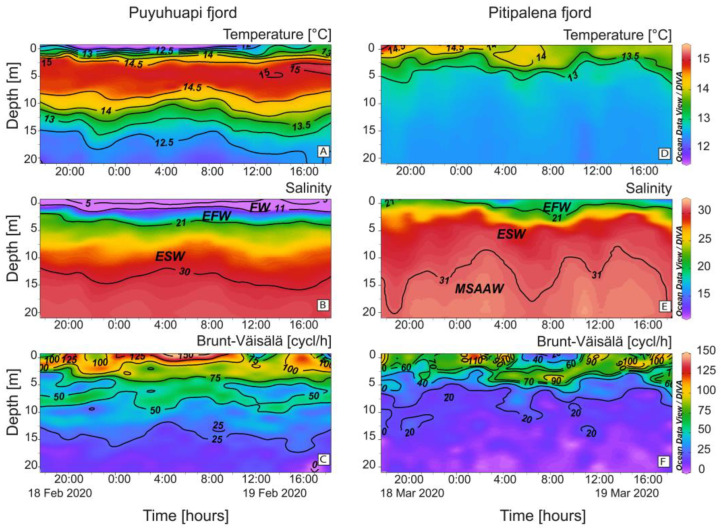
Vertical distribution (0–20 m) every hour of (**A**,**D**) Temperature (°C); (**B**,**E**) Salinity and (**C**,**F**) Brunt–Väisälä frequency at the fixed sampling station in PUY (left panels) and PIT (right panels) fjords during the 24 h surveys in February and March 2020, respectively. Acronyms for water layers in the fjords stand for: FW (S < 11); EFW = Estuarine Fresh Water (S = 11–21); ESW= Estuarine Salty Water (S = 21–31); MSAAW= Modified Subantarctic water (S > 31) [52].

**Figure 3 marinedrugs-21-00064-f003:**
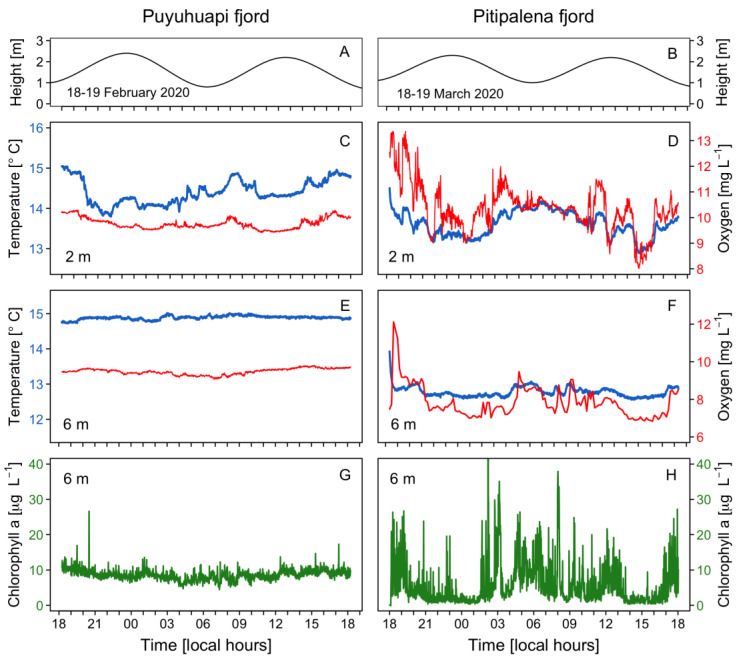
Physical–chemical conditions recorded at the fixed sampling station in PUY (left panels) and PIT (right panels) fjords during 24 h, in February and March 2020, respectively. Tidal amplitude (**A**,**B**); Temperature (blue line) and dissolved oxygen (red line) measurements at 2 m (**C**,**D**) and 6 m depth (**E**,**F**); Chlorophyll-a recorded at 6 m (**G**,**H**).

**Figure 4 marinedrugs-21-00064-f004:**
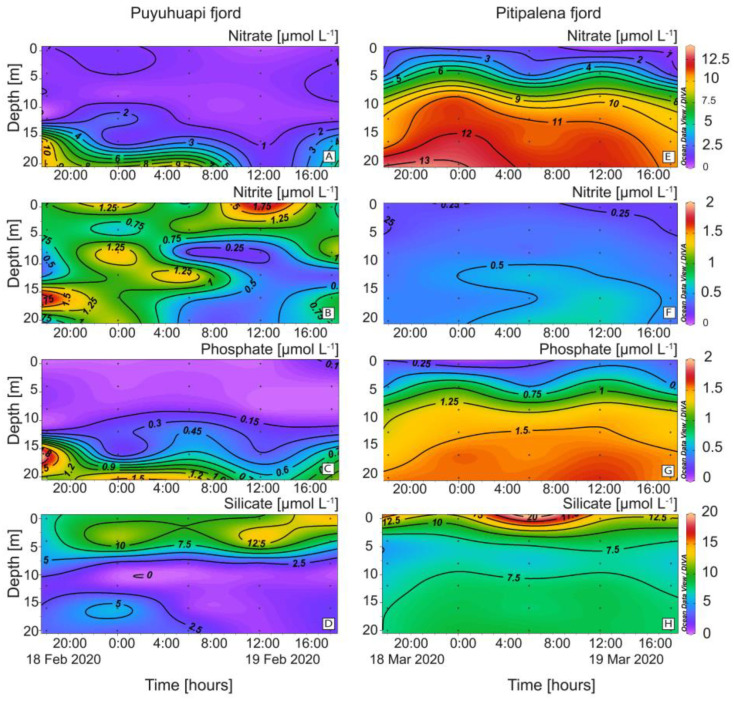
Vertical distribution (every 6 h) of dissolved nutrients at the fixed station in PUY (left panels) and PIT (right panels) fjords during 24 h sampling in February and March 2020, respectively. (**A**,**E**) Nitrate; (**B**,**F**) Nitrite; (**C**,**G**) Phosphate; (**D**,**H**) Silicate.

**Figure 5 marinedrugs-21-00064-f005:**
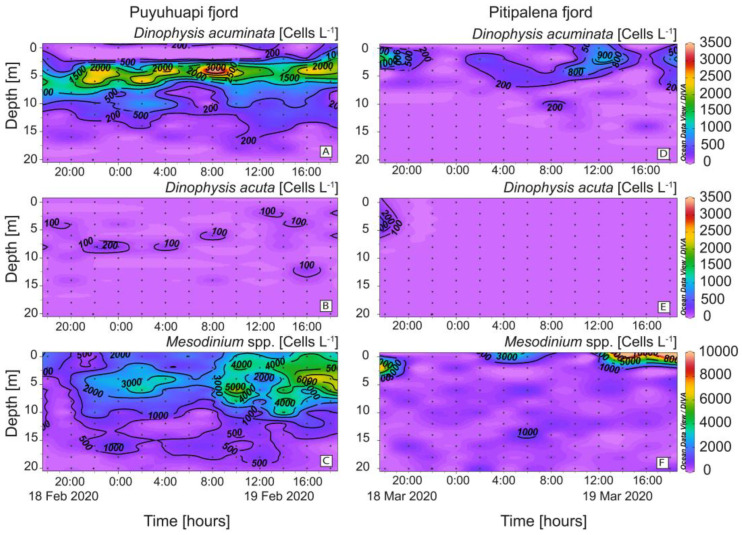
Vertical distribution (every 2 h) of *Dinophysis* and prey at the fixed station in PUY (left panels) and PIT (right panels) fjords during the 24 h studies in February and March 2020, respectively. (**A**,**D**) *Dinophysis acuminata*; (**B**,**E**) *Dinophysis acuta*; (**C,F**) *Mesodinium* spp. Note the different scales between *Dinophysis* and *Mesodinium* panels.

**Figure 6 marinedrugs-21-00064-f006:**
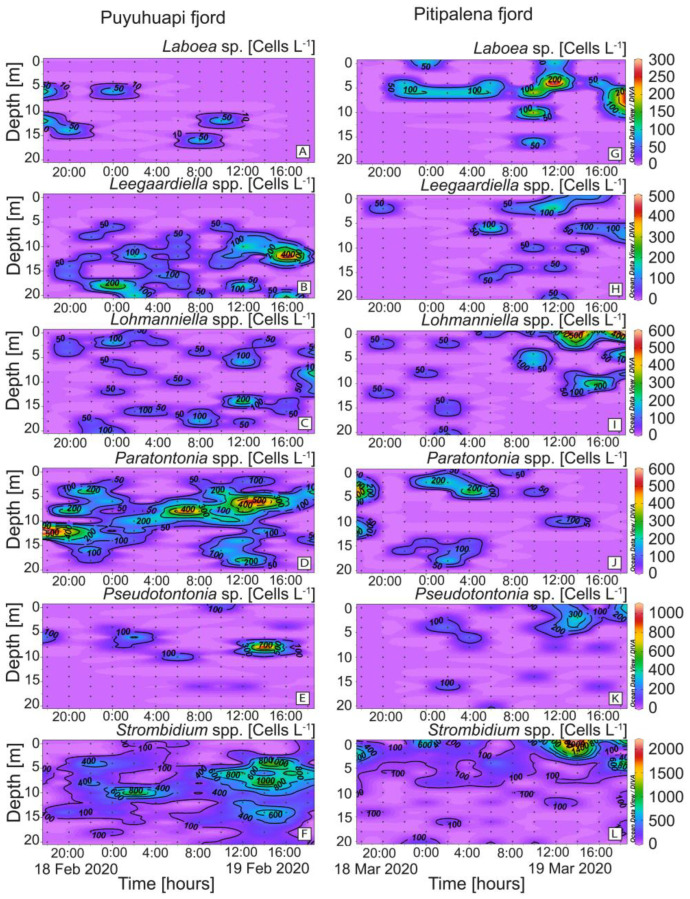
Vertical distribution (every 2 h) of potential plastid-bearing micro-ciliate prey for *Dinophysis* species: (**A,G**) *Laboea* sp.; (**B,H**) *Leegaardiella* spp.; (**C,I**) *Lohmanniella* sp.; (**D,J**) *Paratontonia* spp.; (**E,K**) *Pseudotontonia* sp.; (**F,L**) *Strombidium* spp. at the fixed station in PUY (left panels) and PIT (right panels) fjords during the 24 h studies in February and March 2020, respectively. Note the different scales between panels.

**Figure 7 marinedrugs-21-00064-f007:**
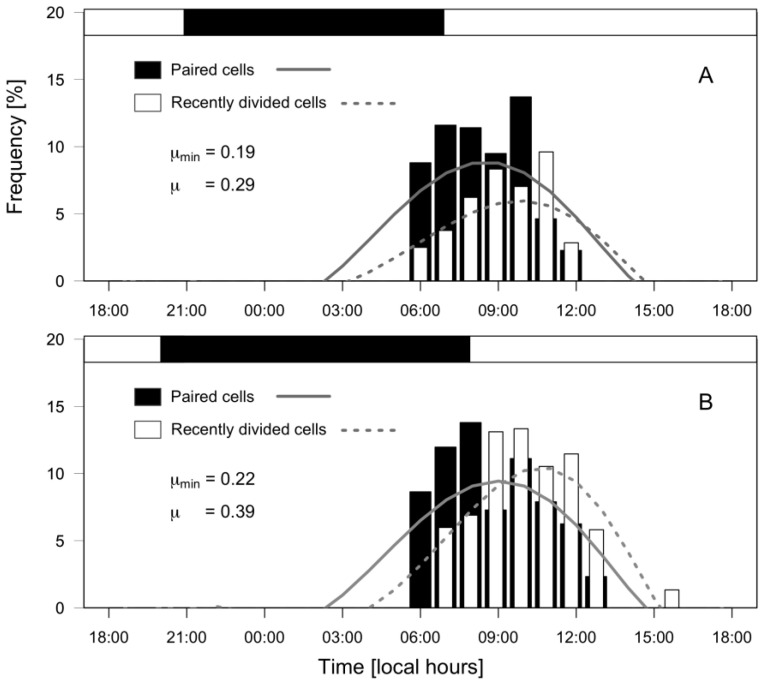
Distribution of frequencies (%) of paired (dividing, black bars) and recently divided (white bars) cells of *D. acuminata* during 24 h monitoring of cell cycle stages at PUY (**A**) and PIT (**B**) fjords. Top horizontal bar indicates the dark hours (between sunset and sunrise).

**Figure 8 marinedrugs-21-00064-f008:**
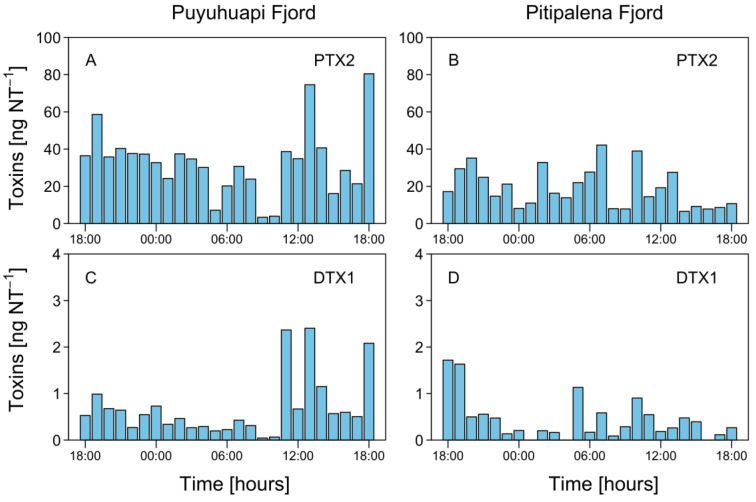
Hourly distribution of lipophilic toxins in vertical (0–20 m) plankton net (20 µm) tows (NT) at the fixed station in PUY (**A,C**), and PIT (**B,D**) fjords during the 24 h studies in February and March 2020, respectively.

**Figure 9 marinedrugs-21-00064-f009:**
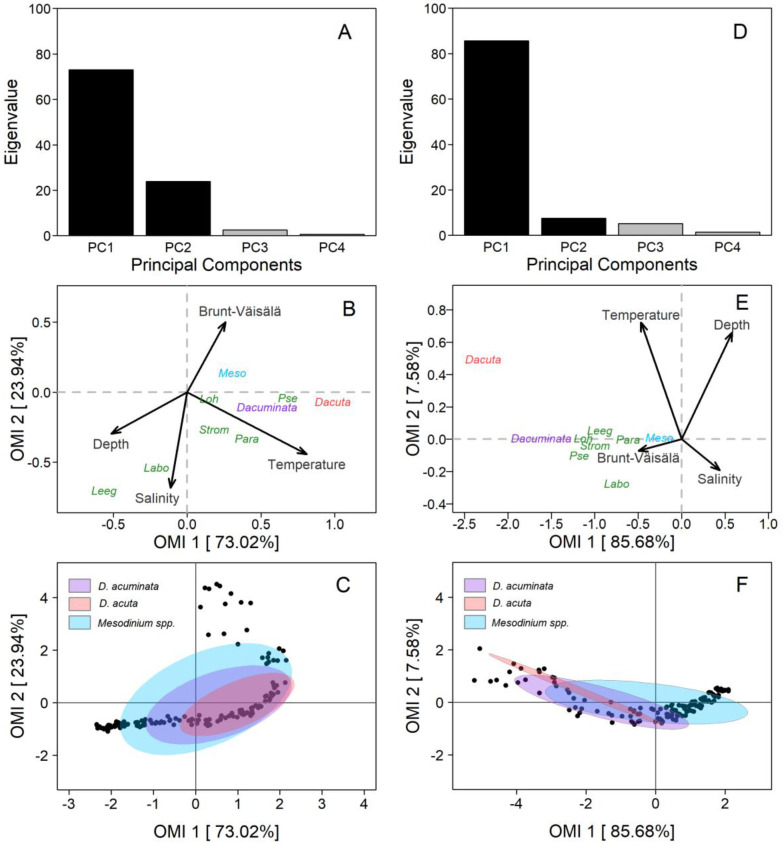
Outlying Mean Index (OMI) analysis of *Dinophysis acuminata*, *Dinophysis acuta*, and co-occurring plastid-bearing micro-ciliates at the fixed station in PUY (left panels) and PIT (right panels) in 18–19 February and 18–19 March 2020, respectively. (**A,D**) Bar plot of the eigenvalue in percentages of the total sum. Black bars are the chosen factorial axis PC1 (OMI1) and PC2 (OMI2); (**B,E**) Species’ realized niche positions of *D. acuminata* (purple), *D. acuta* (pink), their putative prey *Mesodinium* spp. (light blue), and other plastid-bearing ciliates (dark blue) on OMI1 and OMI2 with the canonical weights of environmental variables (see Table 1 for codes); (**C,F**) Realized niche breadth of *D. acuminata*, *D. acuta*, and *Mesodinium* spp. within the environmental space. Black dots represent the samples.

**Figure 10 marinedrugs-21-00064-f010:**
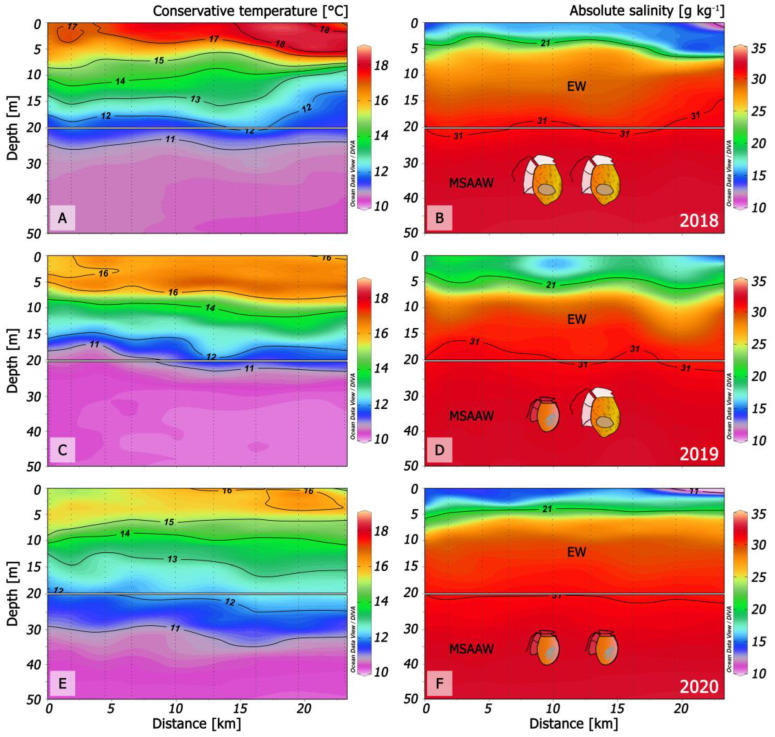
Changes in thermal (left panels) and haline (right panels) structure in three different scenarios in Puyuhuapi fjord (transects from Magdalena sound and across the fjord) that promoted: (**A**,**B**): A record density bloom of *D. acuta* in 2018 [32]; (**C**,**D**): Co-occurrence of *D. acuminata* and *D. acuta* in time but with niche partitioning [8]; (**E**,**F**): Dominance of *D. acuminata* in 2020 (this work).

**Table 1 marinedrugs-21-00064-t001:** Niche parameters estimated with the Outlying Mean Index (OMI) analysis for *Dinophysis acuminata*, *D. acuta,* and their micro-ciliate potential prey at a fixed sampling station in Puyuhuapi (18–19 February 2020) and Pitipalena (18–19 March 2020) fjords during intensive 24 h samplings. Niche parameters are given as absolute values for each species. Inertia (total variability), OMI (marginality), Tol (Tolerance), Rtol (Residual Tolerance). *p*-values were calculated with 10,000 random permutations that yielded a higher value than the observed marginality (OMI). Bold values were significant (*p* < 0.05).

Species	Code	Inertia	OMI	Tol	Rtol	*p*-Value
**Puyuhuapi Fjord**						
*Dinophysis acuminata*	Dacuminata	2.77	0.30	1.15	1.32	**<0.01**
*Dinophysis acuta*	Dacuta	2.55	0.97	0.76	0.82	**<0.01**
*Mesodinium* spp.	Meso	4.09	0.07	2.68	1.22	**<0.01**
*Leegaardiella* sp.	Leeg	2.23	0.81	0.73	0.68	**<0.01**
*Paratontonia* spp.	Para	2.05	0.27	0.46	1.32	**<0.01**
*Strombidium* spp.	Strom	2.81	0.10	0.61	2.09	**<0.01**
*Pseudotontonia* sp.	Pse	2.88	0.47	0.95	1.45	0.18
*Laboea* sp.	Labo	1.70	0.33	0.28	1.09	0.59
*Lohmanniella* sp.	Loh	3.61	0.02	1.52	2.06	0.81
**Pitipalena Fjord**						
*Dinophysis acuminata*	Dacuminata	6.39	2.43	2.78	1.18	**<0.01**
*Dinophysis acuta*	Dacuta	9.27	5.59	1.23	0.46	**0.01**
*Mesodinium* spp.	Meso	4.57	0.07	3.83	0.67	**<0.01**
*Leegaardiella* sp.	Leeg	2.23	0.81	0.73	0.68	**0.02**
*Paratontonia* spp.	Para	4.38	0.39	3.18	0.80	0.09
*Strombidium* spp.	Strom	5.76	1.04	3.78	0.95	**<0.01**
*Pseudotontonia* sp.	Pse	6.09	1.41	3.51	1.17	**<0.01**
*Laboea* sp.	Labo	3.49	0.67	1.50	1.32	0.50
*Lohmanniella* sp.	Loh	6.69	1.31	4.42	0.95	**<0.01**

**Table 2 marinedrugs-21-00064-t002:** PERMANOVA analysis based on Euclidean method with the environmental variables explaining the cell densities of *D. acuminata* and *D. acuta* at a fixed sampling station in Puyuhuapi (18–19 February 2020) and Pitipalena (18–19 March 2020) fjords during 24 h intensive sampling. *p*-values were calculated with 10,000 random permutations. Bold values were significant (*p* < 0.05).

Predictive Variables	Df	SS	R^2^	Pseudo-F	*Pr > F*
**Puyuhuapi Fjord**					
Depth	1	0.31	0.0056	0.0684	0.92
Temperature	1	15.80	0.0289	3.5404	**0.03**
Salinity	1	0.20	0.0004	0.0453	0.95
Brunt–Väisälä frequency	1	0.60	0.0011	0.1334	0.85
Residuals	99	441.68	0.8098		
Total	103	545.41	1.0000		
**Pitipalena Fjord**					
Depth	1	1.05	0.0070	0.3531	0.67
Temperature	1	17.64	0.1172	5.9359	**0.01**
Salinity	1	11.26	0.0748	3.7880	**0.03**
Brunt–Väisälä frequency	1	1.89	0.0126	0.6385	0.51
Residuals	43	127.81	0.8488		
Total	47	150.58	1.0000		

**Table 3 marinedrugs-21-00064-t003:** Detection (LOD) and quantification (LOQ) limits for the toxins detected in the analyses.

Toxin	LOQ (ng/mL)	LOD (ng/mL)
OA	0.50 ± 0.03	0.32 ± 0.03
DTX1	0.39 ± 0.02	0.24 ± 0.02
PTX2	4.33 ± 0.25	2.61 ± 0.25

## Data Availability

Not applicable.

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
