# Peer review of "Dinophysis acuminata or Dinophysis acuta: What Makes the Difference in Highly Stratified Fjords?"

_marinedrugs, 2023, doi:10.3390/md21020064_

Round 1

Reviewer 1 Report

Overall, mostly well written text (until the Discussion-- which requires some editing).

D. acuta is said to be the only OA producer, but there is no data for OA or any comments re: OA  in the results – until line 440 in Discussion. 

So if the goal of this paper is meant to be a forecast of D. acuta, should there be some OA when it was observed?? (The Conclusion implies only D. acuta is a concern; D. acuminata  is not a concern, as stated in the Abstract?)

Abstract

line 38: OMI is not defined; should add “Outlying Mean Index analysis (OMI).

Note: if this acronym was used to meet a word limit, it seems unnecessary since line 46 has plenty of space remaining).

Lin41:  this sentence is awkward:  to provide short-term forecasts of no risk of D. acuta events.

Suggest revising to:

 These physical conditions may be useful in providing a short-term forecast when there is no risk of a D. acuta bloom.

Line 93—end of this paragraph-- authors should state something re: relative concern of OA vs. DTX1 and PTX and human health. Why do we care if event is D acuta or D acuminata??

Line 108  “proved” not probed

Line 174 should be revised, as it seems PO4 is similar to NITRATE (not nitrite) and PO4 is ~2x higher at PIT, not PUY

Line 175, it looks like both sites showed harmonic changes—one up; one down

177 is “phosphatecline” a correct term?

179 is “silicatecline” sa correct term?

Line 199 add (Fig. 3 E&F) here since you refer to DO

Discussion

Line 334—345- This first paragraph is a repeat of Results; so suggest removing this paragraph, or moving some of the text, to conclusion (which reads as simple bullet points)??

Line 351 remove “the” before two

Section 3.1 (358-428)  should  be shortened and improved by focusing this discussion on how data support  objectives of the study

Line 359. Remove “;” and start a new sentence “There …

Line 381  18o  (replace a with o)

Line 393: add space “ in summer”

Line 451 “soft” ? how does this describe antapical end?

Line 477; replace suggest with “suggests”

Line 489  replace “it” with D. acuta  (to clarify sentence)

Section 3.4 is unclear; repeats previous information from previous paragraph. I would suggest removing, unless there is a significant point (and it can be made clear)

Section 4.3

Were filtrates for nutrients frozen or analyzed immediately? This (whichever) should be stated.  From comment re: ammonia, it seems they were not analyzed immediately

Conclusion: reads like simple bullet points. Should address how study answered questions and provide more substance (e.g. Abstract—which has very detailed results)

Figures

Add pictures of D. acuminata “ dividing” and “recently divided” to Fig. 7  (the Reguera 2003 paper only has images for D. caudata

Author Response

We recognize the constructive criticism and the detailed reading and editing made by reviewer 1

R1; Overall, mostly well written text (until the Discussion-- which requires some editing). D. acuta is said to be the only OA producer, but there is no data for OA or any comments re: OA in the results – until line 440 in Discussion. 

 AUTHORS: In the original Introduction (L 84-94) we wrote: “differences in toxin profiles asociated with blooms of D acuta (OA, DTX and PTX2) and D. acuminata (only PTX2 in most cases and sometimes PTX2 and DTX1 but no OA) in Southern Chile”. We have made major changes in the Introduction and discussion (section 3.2) to explain this issue more clearly.

In the present work OA was not detected at all in any net tow samples from Pitipalena or Puyuhuapi Fjords. This observation coincided with the absence of D acuta in PIT and just detection levels in PUY. We thought this supported our view (from previous cruises) that D. acuta is the only producer of OA in Chile and this toxin presence/absence can be used as a tracer of D.acuta

R1:So if the goal of this paper is meant to be a forecast of D. acuta, should there be some OA when it was observed?? (The Conclusion implies only D. acuta is a concern; D. acuminata is not a concern, as stated in the Abstract?)

AUTHORS: The main goal of this paper was to identify microscale hydrographic features that promote bloom development of each species of Dinophysis. The “physical niche”. This is crucial if  we want to improve early warning of the toxic outbreaks with operational oceanography tools. And second, the toxins associated with their blooms. Our resulta confirm that although the most frequent blooms of D acuminata have been associated with only PTXs (now deregulated!) there are also strains of D acuminata producing DSP toxins (in this case DTX1) in North Patagonia that pose a risk for public health and may lead to harvesting ban.

Abstract

R1:line 38: OMI is not defined; should add “Outlying Mean Index analysis (OMI). Note: if this acronym was used to meet a word limit, it seems unnecessary since line 46 has plenty of space remaining). DONE

R1:Lin41: this sentence is awkward: to provide short-term forecasts of no risk of D. acuta events. Suggest revising to: These physical conditions may be useful in providing a short-term forecast when there is no risk of a D. acuta bloom.

AUTHORS: We have used the above sentence suggested by the reviewer:

R1: Line 93—end of this paragraph-- authors should state something re: relative concern of OA vs. DTX1 and PTX and human health. Why do we care if event is D acuta or D acuminata??

AUTHORS: We have added new text in introduction and in discussion. We hope the message is now clearer.

R1: Line 108  “proved” not probed. CORRECTED

R1: Line 174 should be revised, as it seems PO4 is similar to NITRATE (not nitrite) and PO4 is ~2x higher at PIT, not PUY

Line 175, it looks like both sites showed harmonic changes—one up; one down

AUTHORS:  Correct. The two points have been addressed and rephrased

R1:177 is “phosphatecline” a correct term?

179 is “silicatecline” a correct term?

 AUTHORS. These two terms have been corrected to “Phosphocline” and “silicaclines”, the correct terms used by chemical oceanographers,

R1: Line 199 add (Fig. 3 E&F) here since you refer to DO. DONE

Discussion

R1:This first paragraph is a repeat of Results; so suggest removing this paragraph, or moving some of the text, to conclusion (which reads as simple bullet points)??

AUTHORS: It has been removed and a new introductory para. to discussion has been written.

R1: Line 351 remove “the” before two. DONE

R1: Section 3.1 (358-428) should be shortened and improved by focusing this discussion on how data support objectives of the study DONE

R1:Remove “;” and start a new sentence “There …DONE

R1: Line 381  18o  (replace a with o) DONE

R1: Line 393: add space “ in summer”         DONE

R1: Line 451 “soft” ? how does this describe antapical end?

AUTHORS

The word “soft” was removed. It referred to a “soft texture of the hypothecal plates” but it is not necessary to explain this here.

R1:Line 477; replace suggest with “suggests” DONE

R1:Line 489  replace “it” with D. acuta (to clarify sentence). DONE

AUTHORS: It has been replaced

R1:Section 3.4 is unclear; repeats previous information from previous paragraph. I would suggest removing, unless there is a significant point (and it can be made clear)

AUTHORS :Section 3.3 and 3.4 have been combined (only 4.3 now). More concise and going to the point (we hope). The message that D. acuminata is tolerant (can survive under a wide rango of T,S …conditions) but at the same time marginal (optimal growth in a very narrow rango) is very important and is original. This conclusion is reached by  observing distributions of D acuminata with depth (this work, 2 previous ones by the same authors and bibliography review) and by a niche analysis (outlying mean index = OMI method). In the same line, D. acuta is found to be much less tolerant

R1: Were filtrates for nutrients frozen or analyzed immediately? This (whichever) should be stated. From comment re: ammonia, it seems they were not analyzed immediately

AUTHORS: The precise procedure followed, step by step, has been detailed in M&M, section 4.3

R1: Conclusion: reads like simple bullet points. Should address how study answered questions and provide more substance (e.g. Abstract—which has very detailed results)

AUTHORS: Conclusions have been re-written in a more explicative tone

R1: Figures

Add pictures of D. acuminata “ dividing” and “recently divided” to Fig. 7 (the Reguera 2003 paper only has images for D. caudata

AUTHORS: This is not correct. In Reguera et al, MEPS 2003, there are pictures of dividing cells of D caudata in Fig 3 and in Fig 14 there are pictures of D acuminata cells stained with DAPI before and after nuclear division and also during and after cytokinesis (paired cells with incomplete development of the sulcal lists. We have cited these images in the text of the submitted manuscript.

Thanks for your dedication!

Reviewer 2 Report

In this study, the data presented by the authors are original and interesting. The authors conducted careful work that may attract the attention of a wide range of specialists focused on Dinophysis acuminata and D. acuta.The paper can be published.

Author Response

R2: Comments and Suggestions for Authors

In this study, the data presented by the authors are original and interesting. The authors conducted careful work that may attract the attention of a wide range of specialists focused on Dinophysis acuminata and D. acuta. The paper can be published.

AUTHORS: We appreciate that you consider our paper of interest for a wide range of specialists and acceptable to be published in Marine Drugs special issue.

We have addressed all the details raise by Reviewers 1 and 3.

Reviewer 3 Report

This is a nice manuscript. I really enjoyed the introduction as I thought it a well written overview of the situation and provides an excellent introduction to the theme and the specifics of the study requirements. The results were well presented, with effective use of distribution figures. For me I would need to see more evidence that the toxin method used is fit for purpose, to help confirm the accuracy of the toxin concentrations determined.

Specific

·         L36 – units for 21-31 and 11-21

·         L38 onwards – please provide full words for acronyms first time of use and avoid use of coded terms that cannot be understood in a stand-alone abstract

·         L86 – “reviewed in 26” rather than “revised”

·         L86 – again, please check that toxins and other acronyms are spelled out in full first time of use e.g. “okadaic acid (OA) in addition to dinophysistoxin 1 (DTX1) and pectenotoxin 2 (PTX2)”. Please check this throughout the manuscript

·         Figure 1, for me it would be most helpful to incorporate an additional map at wider zoom, showing where these regions lie in relation to the rest of the Chilean coastline, if this can be fitted in somehow

·         L131 – ensure units are stipulated for all measurements recorded

·         L239 – another example of an acronym that needs explanation (LC-HRMS), and two toxins (DTX1 and PTX2 that should have been spelled out earlier – so acronyms can now be used

·         L241 – given that quantitation was performed (according to the methods section) by comparing peak areas against those generated from dilutions of traceable, certified CRMs, please make some mention here that the chromatographic retention times specified are those expected following analysis of toxin standards. This would give the reader extra confidence that the toxins are indeed the analytes being described.

·         General question here is why LC-HRMS is used when LC-MS/MS methods are so well established, and are perfect for sensitive, accurate determination of low toxin concentrations. Conversely, whilst the technique is appropriate, to my knowledge there are no validated methods for LTs that utilise HRMS confirmation and quantitation – it seems this brings in an unnecessary complication which (whilst providing additional confirmation from accurate mass detection) leaves me with a little bit of a sense of doubt about the validity of the concentrations described. It would help therefore if there was information presented somewhere in the manuscript which describes method performance characteristics of the LC-HRMS method. Including, LOD/LOQ, linearity range (how do we know the HRMS detector is linear and therefore reliable at the concentrations determined?), accuracy and precision in some form. This could be easily determined and presented along with the chromatograms in the Appendix

·         L357 – should the question marks be where they are – reads a little oddly

·         L439 – OA absence being discussed – again this shows the importance of understanding LC-HRMS method performance especially sensitivity of analysis (LOD/LOQ). The reader needs to know if the absence is method related or actually a real indication of the environmental conditions

·          L656 – subscript font for the “4” in NH4OH

Author Response

This is a nice manuscript. I really enjoyed the introduction as I thought it a well written overview of the situation and provides an excellent introduction to the theme and the specifics of the study requirements. The results were well presented, with effective use of distribution figures. For me I would need to see more evidence that the toxin method used is fit for purpose, to help confirm the accuracy of the toxin concentrations determined.

AUTHORS: Thanks for your encouraging comments. We agree that the discussion could be substantially improved and it has been reduced and reorganized. Concerning the toxin analysis methods, calibration curves, and detection/quantification limits are now provided in text and in supplementary material repectively.

Specific

R3: L36 – units for 21-31 and 11-21

AUTHORS: We have been told by physical oceanographers/modelers colleagues that “Practical salinity should be expressed by dimensionless number only and should be written as, e.g. S = 35.034” It was our mistake not to add “S = “ before the absolute value to provide data with more clarity to the readers. We have added “S” before any salinity value in the manuscript (Reference about this issue: https://salinometry.com/pss-78/)

R3: L38 onwards – please provide full words for acronyms first time of use and avoid use of coded terms that cannot be understood in a stand-alone abstract

AUTHORS:  full words have been provided for Outlying Mean Index (OMI), okadic acid (OA), pectenotoxin 2 (PTX2), dinophysistoxin 1 (DTX1) and diarrhetic shellfish poisoning (DSP)

R3: L86 – “reviewed in 26” rather than “revised” DONE

R3: L86 – again, please check that toxins and other acronyms are spelled out in full first time of use e.g. “okadaic acid (OA) in addition to dinophysistoxin 1 (DTX1) and pectenotoxin 2 (PTX2)”. Please check this throughout the manuscript

AUTHORS: Full words have been added before their acronyms

R3: Figure 1, for me it would be most helpful to incorporate an additional map at wider zoom, showing where these regions lie in relation to the rest of the Chilean coastline, if this can be fitted in somehow

AUTHORS: Fig 1A, the panel showing the whole Chilean coast, has been enlarged and name of the Patagonian provinces added.

R3: L131 – ensure units are stipulated for all measurements recorded

AUTHORS: We used S (for salinity) before salinity values with no units. We corrected the term used to refer to the Brunt Vaisala frequency. “Static stability has been changed to “buoyancy frequency”. Some authors use “s-1” as the unit for buoyance frequency measurements. We have used “cycles h-1” , unit provided by using the CTD software provided by the manufacturers and depicted using the Ocean Data View, version 5.1 (Ocean Data View)

R3: L239 – another example of an acronym that needs explanation (LC-HRMS), and two toxins (DTX1 and PTX2 that should have been spelled out earlier – so acronyms can now be used

AUTHORS: The acronym LC-HRMS is now preceded by the full words it stands for. Unnecessarily repeated full words for DTX1 annd PTX2 have been deleted.

R3: L241 – given that quantitation was performed (according to the methods section) by comparing peak areas against those generated from dilutions of traceable, certified CRMs, please make some mention here that the chromatographic retention times specified are those expected following analysis of toxin standards. This would give the reader extra confidence that the toxins are indeed the analytes being described.

AUTHORS: In traditional chromatography, retention times are important provided that UV, FLD, IR and other detectors are available. In high resolution mass spectrometry, the precise mass value of the detected compound and its fragmentation spectrum are also important parameters in addition to retention time. In this context, new graphs of chromatograms (selected samples and the toxin standards) related to DTX1 and PTX2 have been added In supplementary material. Toxins and standards show the same retention time and a confirmatory spectrum from a fragmentation of the original compound

R3: General question here is why LC-HRMS is used when LC-MS/MS methods are so well established, and are perfect for sensitive, accurate determination of low toxin concentrations. Conversely, whilst the technique is appropriate, to my knowledge there are no validated methods for LTs that utilise HRMS confirmation and quantitation – it seems this brings in an unnecessary complication which (whilst providing additional confirmation from accurate mass detection) leaves me with a little bit of a sense of doubt about the validity of the concentrations described.

AUTHORS: You are absolutely right. Toxin analysis by LC MS/MS with triple quadrupole detectors is a worldwide known analytical method for shellfish toxins. Furthermore, it is the standard technique adopted by the official laboratories in charge of toxin monitoring. Nevertheless, we are not a food quality control lab but a small research centre with an LC-HRMS equipment which is used for different purposes and kinds of analyses, including marine toxins in plankton and shellfish, for research purposes.

R3: It would help therefore if there was information presented somewhere in the manuscript which describes method performance characteristics of the LC-HRMS method. Including, LOD/LOQ, linearity range (how do we know the HRMS detector is linear and therefore reliable at the concentrations determined?), accuracy and precision in some form. This could be easily determined and presented along with the chromatograms in the Appendix

AUTHORS: We have included the calibration curves for OA, DTX1 and PTX2. We also show the LOD and LOQ estimates for each toxin and explain in Material & Methods how they were estimated.

R3: L357 – should the question marks be where they are – reads a little oddly

AUTHORS: The question marks are a mistake. We forgot to delete them for the clean final version. We change the word “filter??” to “promote”

R3: L439 – OA absence being discussed – again this shows the importance of understanding LC-HRMS method performance especially sensitivity of analysis (LOD/LOQ). The reader needs to know if the absence is method related or actually a real indication of the environmental conditions

AUTHORS: These concerns have been addressed in previous sections of this “answer to reviewer 3”

R3:  L656 – subscript font for the “4” in NH4OH. DONE
